# Characteristics of Collagen Changes in Small Intestine Anastomoses Induced by High-Frequency Electric Field Welding

**DOI:** 10.3390/biom12111683

**Published:** 2022-11-13

**Authors:** Caihui Zhu, Li Yin, Jianzhi Xu, Xingjian Yang, Hao Wang, Xiaowei Xiang, Haotian Liu, Kefu Liu

**Affiliations:** 1School of Information Science and Technology, Fudan University, Shanghai 200433, China; 2Academy for Engineering & Technology, Fudan University, Shanghai 200433, China

**Keywords:** small intestine, tissue welding, Raman spectroscopy, orientation analysis

## Abstract

High-frequency electric field welding-induced tissue fusion has been explored as an advanced surgical method for intestinal anastomoses; however, intrinsic mechanisms remain unclear. The aim of this study was to investigate microcosmic changes of collagen within the fusion area, with various parameters. Ex vivo small intestine was fused with mucosa–mucosa. Four levels of compressive pressure (100 kPa, 150 kPa, 200 kPa, 250 kPa) were applied for 10 s in order to fuse the colons under a power level of 140 W. Then, collagen fibers of the fusion area were examined by fibrillar collagen alignment and TEM. Three levels of power (90 W, 110 W, 140 W) and three levels of time (5 s, 10 s, 20 s) were applied in order to fuse colons at 250 kPa, and then collagen within the fusion area was examined by Raman spectroscopy. Fibrillar collagen alignment analysis showed that with the increase in compression pressure, alignment of the collagen in the fusion area gradually increased, and the arrangement of collagen fibers tended to be consistent, which was conducive to the adhesion of collagen fibers. TEM showed that pressure changed the distribution and morphology of collagen fibers. Raman spectroscopy showed that increased power and time within a certain range contributed to collagen cross linking. Peak positions of amide I band and amide III band changed. These results suggested that higher power and a longer amount of time resulted in a decrease in non-reducible cross links and an increase in reducible cross links. Compression pressure, power, and time can affect the state of collagen, but the mechanisms are different. Compressive pressure affected the state of collagen by changing its orientation; power and time denatured collagen by increasing temperature and improved the reducible cross linking of collagen to promote tissue fusion.

## 1. Introduction

High-frequency electric field welding (HFEW) induced by heat is a routine surgical procedure. The feasibility of the technology for induced small intestine anastomosis has been confirmed by previous ex vivo and in vivo experiments [1,2]. Tissue fusion is achieved by application of an alternating high frequency electric field for seconds, which creates soft tissue fusion [3]. In surgery, blood vessel sealing has been clinically approved by RF bipolar vessel sealer (Covidien, Missouri, TX, USA) [4]. Much research efforts have recently been conducted to explore new fusion instruments for small intestinal anastomosis or sealing [5,6,7] which may replace traditional hand suturing or stapling. HFEW can achieve high-quality rapid tissue welding without leaving any foreign materials; thus, it is expected to greatly reduce morbidity, mortality and cost.

The mechanism of HFEW, particularly for blood vessels and small intestine, has been discussed in much of the literature [8,9,10]. Energy-based tissue welding is believed to be the result of simultaneously applied compression pressure (CP) and energy [11,12]. Appropriate CP is a prerequisite for the success of welding, and precise control of energy is key to the quality of welding. Collagen bonds within the fusion area are believed to be pivotal to HFEW. It is widely accepted that energy denatured collagen to a gel-like amalgam, which can form strong bonds within the fusion area [10]. However, the existing literatures and studies have mainly been focused on either direct microscopic observation or mechanical strength testing, including burst pressure and tensile strength [3,13]. The lack of understanding of how CP and energy influence the state of collagen at the molecular level has left several important questions in this field unresolved. Critically, how the collagen changes within the fusion area during the operation, as well as the precise mechanism for the formation of fusion, are still unclear. Answering these questions will provide better understanding and knowledge of the intrinsic mechanisms for HFEW, and also help to develop more effective control methods for HFEW procedures.

Fibrillar collagens are one of the prominent extracellular matrix components. Studies have shown that specific changes in collagen alignment play an important role in contributing to the mechanical properties of tissues [14,15,16]. It is widely accepted that image analysis methods suitable for quantifying fibrillar collagen alignment and orientation are based on intensity derivatives, intensity variation or directional filters [17]. Raman spectroscopy has been extensively used for studying proteins regarding the amino acids, amide bonds between them, and their tertiary structure [18]. Raman spectroscopy provides an attractive way to identify changes within collagen during thermal denaturing without destroying or altering the tissues. By scanning across the fusion area, the individual Raman spectrum at each acquisition point can be combined to form a Raman map, which is similar to a microscopic image. Raman spectroscopy provides direct insights into tissue constituent and structural changes on the molecular level, exposing spectroscopic evidence for the denaturing and restructuring of collagen cross links post-HFEW fusion.

In this study, we aimed to study collagen changes or denaturation at the molecular level during small intestine fusion induced by HFEW. Image analysis methods and transmission electron microscopy (TEM) were used to investigate the impact of CP on fibrillar collagens in energy-induced small intestine anastomoses. Moreover, Raman spectroscopy was utilized to characterize collagen changes under different power (P) and time (T) conditions.

## 2. Materials and Methods

### 2.1. Preparation of Animal Tissue

Fresh colons were harvested from pigs at slaughterhouses and cut into 50–60 mm segments (Figure 1B). The samples were cleaned in order to eliminate feces, and then immersed in 0.9% saline before being delivered at 0–4 °C to the laboratory for the experiments. All of the prepared small intestine segments were used for HFEW experiments within 24 h of harvest. The conducted research did not require ethical approval because the animal material was taken from a slaughterhouse.

### 2.2. Experimental Settings

The success of HFEW is related to three parameters, namely CP, P, and T (Figure 1A). Four CP parameters (100 kPa, 150 kPa, 200 kPa, 250 kPa) were used to study the effect of CP on orientation analysis of collagen fibers. In order to study the effect of P and T on collagen denaturation, three power parameters (90 W, 110 W, 140 W) and three-time parameters (5 s, 10 s, 20 s) were set during HFEW. EKVZ300(Kiev, Ukraine) was used to provide HF energy. An infrared thermal imaging camera (FOTRIC, Shanghai, China) was used to monitor the temperature distribution on the tissue surface during the process. All experimental conditions were repeated at least ten times. The small intestinal fusion apparatus was a homemade device with copper electrode and a hollow round tube, which was insulated and heat-resistant (Figure 1C,D). The copper electrode was installed on the round tube. The appliance delivered the required pressure to the soft tissue through a compression machine (ZQ-990A-1, Zhiqu) using compressive pressure. The machine can automatically adapt to keep the CP constant during the fusion.

### 2.3. Burst Pressure Measurements

Burst pressure (BP) is defined as the maximum pressure measured by the pressure gauge during infusion. The BP measuring device consisted of three parts, including a peristaltic pump, a pressure sensor, and a porcine bowel to be measured (Figure 1F). The device was connected by three ends of a T-shaped tube. The peristaltic pump pressed water from the flume into the fused intestine at a certain flow rate. The maximum pressure measured by the pressure sensor was the BP of the porcine bowel. In this study, the peristaltic pump BT-100CA was made by Jihpump, with a flow rate from 0.07 to 79 mL/min, and the flow rate was set at 5 mL/min.

### 2.4. Orientation Analysis of Tissue Collagen Fibers

Before orientation analysis, the tissue was stained with Sirius red staining. The prepared intestinal slices were mounted on the stereo microscope and were then visualized and imaged. As previously described, the orientation analysis software used in this study was Curve Align, which has been successfully applied to numerous studies on collagen alignment [17]. This software platform uses two fiber analysis methods, CT-FIRE and curvelets fiber representation (CFR). In this study, CFR was used to analyze tissue sections after Sirius red staining. There are three kinds of alignments calculated in Curve Align: (1) alignment with respect to the horizontal, called the “angle” or “orientation,” ranging from 0 to 180°; (2) alignment with respect to a boundary, called the “relative angle” or “relative orientation,” ranging from 0 to 90°; and (3) alignment of fibers with respect to each other, called the “alignment coefficient,” ranging from 0 to 1. In Curve Align, in order to calculate the alignment coefficient, the angle is first multiplied by 2 to map to the orientation range of [0–2π], and then the alignment coefficient is calculated as the normalized vector sum of orientation vectors or the mean resultant vector length in circular statistics. The alignment coefficient ranges from 0 to 1, with 1 indicating perfectly aligned fibers and smaller values representing more randomly distributed fibers [17].

### 2.5. Transmission Electron Microscopy (TEM)

In order to study the effect of CP on collagen denaturation, unfused samples and fused samples at 250 kPa (a representative CP at which BP is the highest) were examined by TEM to provide a representative assessment of the microscopic effects on collagen. TEM was performed under the guidance of an experienced technician, according to a standard protocol, and the results were viewed with a HITACHI Transmission Electron Microscope (HT7800, Tokyo, Japan).

### 2.6. Raman Micro-Spectroscopy

Raman spectrometer (NTEGRA Spectra-II) was utilized to analyze the fusion area of small intestine. Excitation wavelength was 785 nm. Range of the Raman spectrum was limited to 900–2000 cm^−1^. Firstly, small intestines were fused under different Ps and Ts at 250 kPa, then, Raman spectrum was collected from the fusion area and multiple frequency points were collected for each sample. The average value was taken in order to reduce the influence of signal noise. The Raman spectrum data obtained from samples were edited by Labspec5.0 software and smoothed. Baselines were removed by polynomial fitting. Gauss–Loren function was utilized to calibrate the characteristic peak and the information on characteristic peak position, extreme value.

### 2.7. Statistical Analysis

Statistical analysis was performed using SPSS software ver. 20 (SPSS, IBM, Chicago, CA, USA). The significance level was set at 0.05 for the Kruskal–Wallis test and adjusted according to Bonferroni for post hoc tests. The statistical significance of BP under different Ps and Ts parameters was determined using one-way analysis of variance (ANOVA).

## 3. Results

### 3.1. Orientation Analysis Results

Before the analysis of collagen fibers, the welding quality of the small intestine was evaluated at a macroscopic level. The burst pressure can intuitively reflect the welding quality or welding strength. In this study, there were 50 small intestines fused under different CPs (*n* = 50), Ps (*n* = 20), and Ts (*n* = 20) (Table 1). Burst pressures were measured after the small intestines were fused under different CPs (P = 140 W and T = 10 s). Burst pressure increased with the increase of CP, and the burst pressure were 11.30 ± 6.52 mmHg, 23.40 ± 8.10 mmHg, 27.20 ± 5.29 mmHg, and 35.05 ± 10.28 mmHg, respectively.

The control specimen was a normal small intestine without CP or fusion, which showed collagen distribution in the tissue. The collagen fiber which was dyed red was mainly concentrated in the submucosa and serous membrane layer, with a slight amount of collagen fiber in the mucosa and muscle layer, which have an effect on connected cells and tissues (Figure 2A). After fusion, collagen fibers in the submucosa and serosa remained intact, but a large number of collagen fibers were compressed and aggregated, and the contours of collagen fibers became blurred (Figure 2B). The Sirius red stained images were imported into Curve Align, and the orientation vector distribution of collagen fibers was obtained after CFR treatment. As is shown in Figure 2C, the orientation of collagen fibers in untreated small intestinal tissue was relatively dispersed, and the overall orientation of collagen fibers could not be identified. However, the orientation of collagen fibers in the mucosa on both sides of the treated small intestinal tissue was relatively consistent, and the overall direction was horizontal (Figure 2C,F,I). The angle values of collagen fibers were spaced by 5° and distributed in the range of 0–180°.

Taking the vertical boundary of the image as the reference boundary, we analyzed the angle distribution and angle value of collagen fibers in the fusion area before and after welding under different CP conditions. As shown in Figure 2C–K, with the increase in CP, collagen fibers in the vertical direction gradually decreased, and also tended to be oriented horizontally. At the same time, the holes in the fusion area gradually became smaller, and boundaries of the two tissues became blurred, which indicates that the fusion quality increased. The improvement in fusion quality can be confirmed by the burst pressure results (Table 1). 

The alignment coefficient was able to quantitatively measure the angle distribution of collagen fibers in the region, which ranged from 0 to 1, with 1 indicating perfectly aligned fibers, and smaller values representing more randomly distributed fibers. The average alignment coefficient of unfused tissue was 0.61, which gradually increased with the increase in CP, and approached 1. When CPs were 100 kPa, 150 kPa, 200 kPa and 250 kPa, the alignment coefficient of the fusion area was 0.90 ± 0.0022, 0.96 ± 0.0062, 0.98 ± 0.0054, and 0.99 ± 0.0002, respectively (Figure 2L and Table 1). The alignment coefficient of collagen fibers after fusion was significantly higher than that of the tissue before welding, which also reflected the cross linking of collagen fibers in the fused small intestine during the welding process. 

### 3.2. Transmission Electron Microscopy Results

Transmission electron microscopy (TEM) was used to assess the collagen fibrils of unfused porcine small bowel (Figure 3A,B) and fused small bowel, which was fused at 250 kPa (Figure 3C,D). The control specimen was a normal small intestine without CP or fusion, which showed typical periodic light and dark streaks on fibrils. The borders of collagen fibrils appeared as sharp regular circles, and the background was clean (Figure 3B). By comparison, fused small intestine that underwent fusion with CP of 250 kPa was observed to be grossly fused, and the broken collagen fibrils were randomly diffused in the extracellular matrix (Figure 3C). The borders of the collagen fibrils were fuzzy, and the fibrils were darker and thicker (Figure 3D). In addition, collagen fibers of fused samples lost their periodicity, and the background was blurred. The collagen fibers became dense, and the background was dull. The boundary between transverse-sectioned collagen fibers and longitudinally sectioned collagen fibers became completely fuzzy.

### 3.3. Raman Micro-Spectroscopy Results

There were 60 samples fused at different Ps (*n* = 30) and Ts (*n* = 30) and used for the burst pressure test. Six fused small intestines were imaged using Raman spectroscopy. Seven Raman points per sample were used for analysis. Raman maps were collected from selected regions within welding areas (Figure 2B). The collagen endmember spectrum in each map included all the characteristic features previously reported in Raman spectroscopy studies of collagen and collagen-rich tissues. Specifically, Raman bands corresponding to a C–N stretch of proline (919 cm^−1^), proline (1042 cm^−1^), amide I (1600–1690 cm^−1^), and amide III (1250–1261 cm^−1^) are notable. 

The mean of all spectra collected from the fused areas which underwent fusion at different Ps and Ts were shown in Figure 3. In the fused tissue, a shift in the peak maximum occurred in amide Ⅰ (1600–1690 cm^−1^) and amide III (1252–1270 cm^−1^), and a change in shape was observed at 1443 cm^−1^ (Figure 4A,B). Many peaks, including the 940 cm^−1^ peak representing the protein α helix, did not appear to change peak position or shape. The peak maximum of fused porcine bowel showed changes in the amide III band (1250–1261 cm^−1^) and shifted to a lower wavenumber under different time and power conditions (Table 2). Meanwhile, the peak maximum of the amide Ⅰ band (1600–1690 cm^−1^) in fused porcine bowel showed a shift to a higher wavenumber in all Ps and Ts (Figure 4C,D). Compared with unfused tissue, a higher wavenumber corresponds to a higher burst pressure. The peak maximum of the amide Ⅰ band, corresponding to 90 W (9.41 mmHg) shifted to 1660 cm^−1^; when the power was 140 W (34.08 mmHg), the corresponding peak shifted to 1665 cm^−1^. This result also applied to the time parameter; as the welding time increased, the peak position of amide Ⅰ band moved to a higher position.

The vibrations of amide Ⅰ and amide Ⅲ were the most sensitive to changes in the conformation of secondary structures of collagen. In order to analyze thermal changes in the amide groups for collagen, we compared the Raman intensity ratio of the most intense bands, at 1250 cm^−1^ for amide Ⅲ and 1660 cm^−1^ for amide Ⅰ, with the intensity of the band at 1450 cm^−1^. The band at 1450 cm^−1^ was related to the CH2 banding vibration, which was practically insensitive to the secondary structure. The intensity ratios of the Raman bands characteristic of α-helix frequencies are presented in Figure 3E,F. With the extension of T and the increase in P, the intensity ratios of the Raman bands decreased gradually, proving that the secondary structure of collagen protein was changed.

## 4. Discussion

As a new surgical suture method, HFEW has many advantages, such as the lack of foreign bodies left in the body, low price, and shorter operation time [11,19], and has received much attention from surgeons. It has been demonstrated in some studies that HFEW creates burst pressures as high as those formed by staplers [6], and the technique has been successfully used in in vivo small bowel anastomosis [1,20]. Collagen is widely distributed in small intestinal tissues (Figure 2A) and is also an important raw material for tissue welding. For this reason, we sought to further explore the mechanism of tissue welding technology by studying the influence of different parameters on collagen during welding.

In the study of small intestine sealing or end-to-end anastomosis, burst pressure gradually increased with the gradual increase in the CP [2]. The burst pressure results of fused small intestine under different CPs also support this conclusion (Table 1). When the CP reached around 250 kPa, the burst pressure reached the maximum [13]. Compared with the unfused sample, the fused sample showed thicker denatured collagen fibrils at 250 kPa (Figure 3). Similar findings of thicker collagen fibrils have been observed for the skin after RF treatment [21], which is also apparent for laser vessel welding and RF-induced small bowel ligation [13]. Within a certain range, higher CP can ensure higher burst pressure and reduce the size and number of pores in the fusion. 150–250 kPa is the ideal pressure range, as a successful closure was defined as a burst pressure greater than 15.4 mmHg [19].

We believe that higher CP can provide better conditions for the denaturation and interweaving of collagen fibers. Through collimation coefficient study of collagen fibers, we found that the collimation of collagen fibers improved, and that it tended to be horizontal at higher CP, especially when CP was 250 kPa. The collimation coefficient of collagen fibers is related to the mechanical characteristics of the tissue. The improvement of collimation coefficient is conducive to the interweaving of collagen fibers. Higher collimation coefficient corresponds to higher burst pressure, which is similar to the conclusion obtained by other testing methods [2,9].

The proper range of CP is only a prerequisite for successful fusion, and the denaturation of collagen mainly depends on certain energy input or temperature. Previous reports have suggested that changes in collagen were significant in RF fusion [22,23]. In order to further explore the effect of power and time on fusion quality, we compared the collagen spectra from both the control and fused small intestine tissue under the same CP (250 kPa). The spectrum of the fusion area showed similar band shifts and biomolecular bond changes in two independent samples. These trends included the denaturing of collagen, shown through the significant shift in the peak maximum in the amide Ⅰ (1655 cm^−1^) band to higher wavenumbers in the fused area, which suggested an increase in reducible cross links and a decrease in non-reducible crosslinks within the collagen [24]. Additionally, it has been previously reported that a shift in the 1302 cm^−1^ peak was related to collagen thermal denaturing [25], which shifted to a higher peak, and changed shape in this study. Lastly, the apparent shift in the 1250–1261 cm^−1^ peak’s maximum intensity to lower wavenumbers also implicate that cross links may have been reduced or broken [26]. Fused tissue under different power and time conditions demonstrated several similar changes, including shifts in the 1250–1261 cm^−1^, 1302 cm^−1^ and 1655 cm^−1^, indicating a denaturing of collagen and, more specifically, a decrease in non-reducible cross links and an increase in reducible cross links.

A current passing through the tissue causes a certain rise in temperature, which is the main reason for thermal tissue denaturation. During the welding process, the temperature rises slowly to a certain peak and then fluctuates within a certain range (Figure 5A,B). In this study, a peak temperature between 60 °C and 90 °C can ensure the necessary collagen denaturation in tissue [10,26], which is believed to be essential for strong fusion. Temperatures below 60 °C may not lead to the denaturation of collagens, but excessive temperatures need to be avoided, as they lead to permanent damage of the tissue or necrosis. With the extension of time, the temperature did not increase, but reached the peak, and was then maintained within a certain range. Compared with 5 s and 10 s, the burst pressure was larger at 20 s, which is because the tissue in the fusion area was heated more fully, and the denaturation range was increased. Within a certain range, the increase in power and the prolonging of time leads to the increase in burst pressure; meanwhile, the secondary structure of collagen changes. The vibrations of amide Ⅰ and amide Ⅲ are most sensitive to changes in the conformation of secondary structures of collagen. I (amide Ⅲ)/I (~1450 cm^−1^) and I (amide Ⅰ)/I (~1450 cm^−1^) gradually decreased with the increase in power and the extension of time, which indicated that the secondary structure of collagen was changed.

## 5. Conclusions

As an important biological component of the small intestine, collagen plays a key role in the HFEW process. Studying the changes in collagen in the welding process will help us to better understand the welding mechanism. Tissue welding requires accurate control of two important parameters, pressure and energy, which are determined by both power and time. Pressure and energy have different influence mechanisms on tissue welding. This study explored the mechanism of tissue by studying the effects of various parameters on collagen. The collimation coefficient of collagen fibers is related to the burst pressure of the fused tissues. CP mainly affects collimation of collagen fibers. Within a certain range, with the increase of CP, the collimation coefficient of collagen fiber increases, and the pores at the weld become smaller. Energy (P, T) mainly affects the denaturation of collagen. We studied the denaturation characteristics of collagen using several special Raman spectroscopy peak features. Under different power and time conditions, there were several changes of characteristic peaks shift including 1252–1261 cm^−1^, 1302 cm^−1^, and 1655 cm^−1^, demonstrated a denaturing of collagen and, more specifically, a decrease in non-reducible crosslinks and an increase in reducible crosslinks. Reducible cross links are related to burst pressure. With increasing power and prolongation of time, burst pressure and reducible crosslinks increase. Meanwhile, temperature is the main reason for the change in collagen’s secondary structure, and is related to the cross link of collagen. Under the combined action of P and T, the tissue temperature increases, and I (amide Ⅲ)/I (~1450 cm^−1^) and I (amide I)/I (~1450 cm^−1^) gradually decrease, which means that the secondary structure of collagen is changed.

## Figures and Tables

**Figure 1 biomolecules-12-01683-f001:**
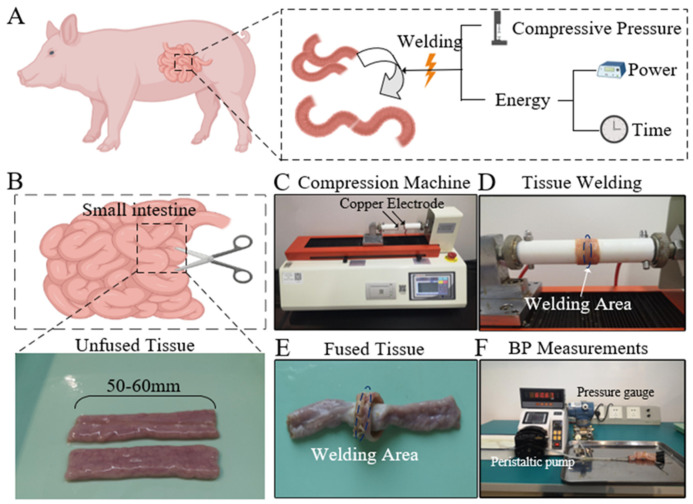
(**A**). Three important parameters in the process of HFEW. (**B**). Small intestine. Fresh small intestine was cut into 50–60 mm segments. (**C**–**E**). Experimental setups for HFEW induced fusions of the porcine bowel. (**F**). Experimental setup for BP measurements.

**Figure 2 biomolecules-12-01683-f002:**
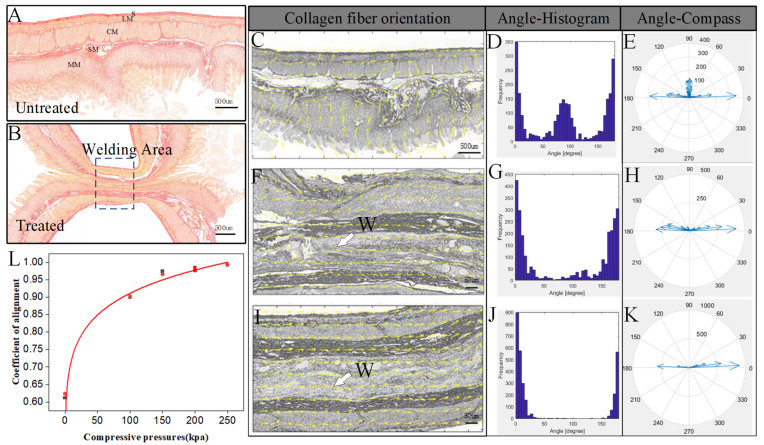
(**A**,**B**). Transverse histological slices of normal and fused small intestine stained with Sirius red. Collagen is dyed red. S, serosa; LM, longitudinal muscle; CM, circular muscle; SM, submucosa; MM, muscularis mucosa. (**C**–**E**). Tissue collagen fiber orientation, angle histogram, and angle compass for unfused small intestine. (**F**–**H**). Tissue collagen fiber orientation, angle histogram, and angle compass for the fused small intestine at 200 kPa and 250 kPa. Coefficient of alignment for different compressive pressure levels. The yellow arrow represents the orientation vector of the fiber. W, welding area (**I**–**L**).

**Figure 3 biomolecules-12-01683-f003:**
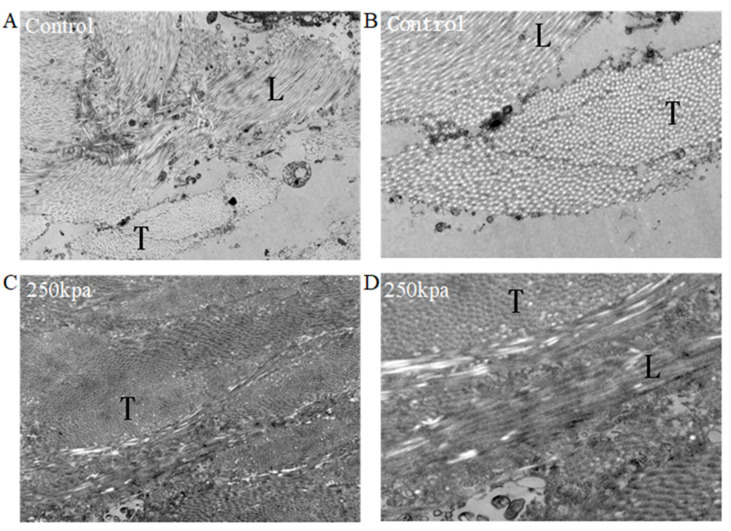
Representative TEM of unfused and fused small intestine. (**A**,**B**) Electron micrograph of control (unfused) small intestine (**A**: magnification 3000×, **B**: magnification 7000×). (**C**,**D**) Electron micrograph of fused small intestine at 250 kPa ((**C**): magnification 3000×, (**D**): magnification 7000×). T, transverse-section of collagen fibrils; L, longitudinal-section of collagen fibrils.

**Figure 4 biomolecules-12-01683-f004:**
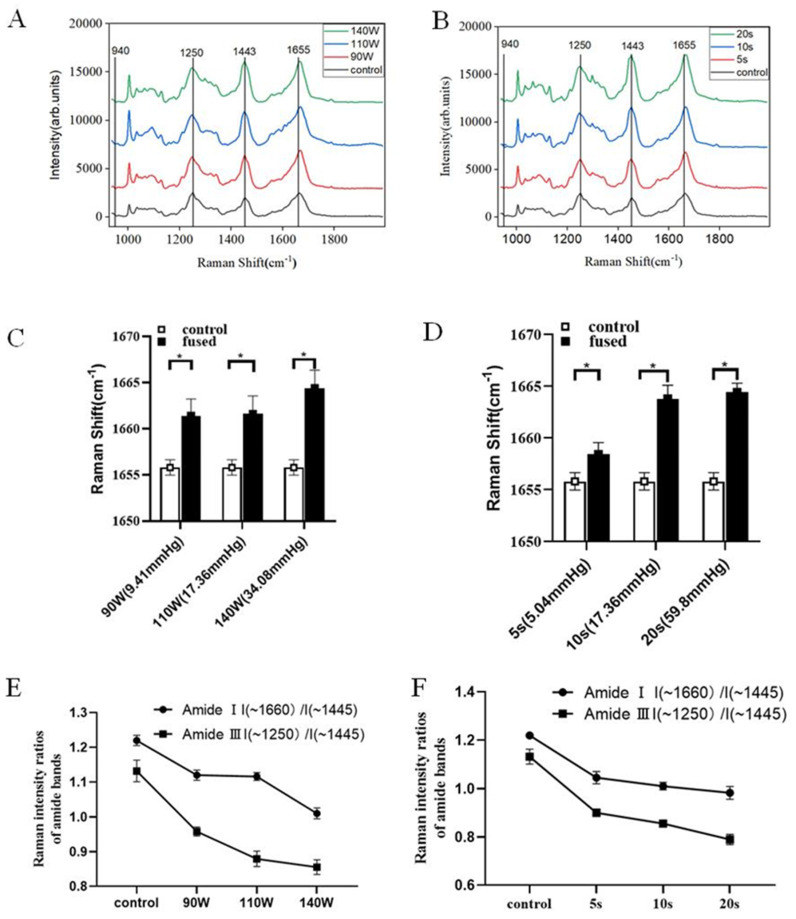
(**A**,**B**). Mean Raman spectra of control and fused areas mapped in porcine bowel tissue undergoing different Ps and Ts. Raman Shifts of 940 cm^−1^, 1250 cm^−1^, 1443 cm^−1^ and 1655 cm^−1^ are highlighted corresponding to the protein alpha helix, amide Ⅲ, CH2 wag, and amide Ⅰ. (**C**,**D**). Peak maximum Raman shift (cm^−1^) location of amide I in small intestine tissue comparing fused tissue and control tissue. Samples were fused with different Ps and Ts. The corresponding average burst pressure is in parentheses, * *p*-value < 0.05. (**E**,**F**). Raman intensity ratios of amide I(~1660 cm^−1^)/I (~1445 cm^−1^) and amide Ⅲ I(~1250 cm^−1^)/I(~1445 cm^−1^) in the small intestine’s fused area, undergoing different Ps and Ts.

**Figure 5 biomolecules-12-01683-f005:**
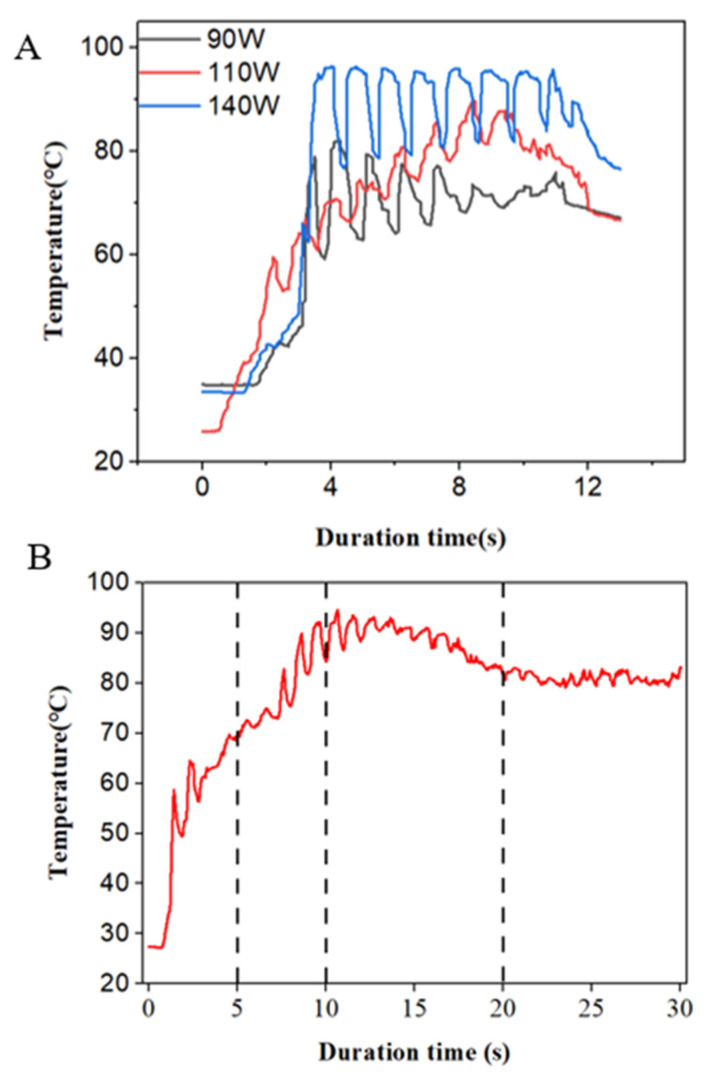
(**A**). Characteristics of tissue temperature at small intestine fusion area under different power. (**B**) Temperature at fusion area changes with time when power is 140 W.

**Table 1 biomolecules-12-01683-t001:** Burst pressure and alignment coefficient of fused small intestines at varying CPs.

Compressive Pressure (kPa)	Burst Pressure (Mean ± SD) (mmHg)	Alignment Coefficient of Collagen Fibers
0	0	0.61 ± 0.0086
100	11.30 ± 6.52	0.90 ± 0.0022
150	23.40 ± 8.10	0.96 ± 0.0062
200	27.20 ± 5.29	0.98 ± 0.0054
250	35.05 ± 10.28	0.99 ± 0.0002

kPa kilopascal, SD standard deviation, mmHg millimeters of mercury.

**Table 2 biomolecules-12-01683-t002:** Raman Spectrum Result.

Sample	Position (~1250 cm^−1^)	Position (~1655 cm^−1^)	I (~1250 cm^−1^)/I (~1445 cm^−1^)	I (~1660 cm^−1^)/I (~1445 cm^−1^)
90 W × 10 s	1250	1655.8	1.132	1.220
110 W × 10 s	1248	1661.4	0.957	1.120
140 W × 10 s	1248	1661.6	0.879	1.116
140 W × 5 s	1246	1664.4	0.855	1.010
140 W × 20 s	1249	1658.4	0.900	1.045
90 W × 10 s	1248	1665	0.789	0.982

## Data Availability

The datasets generated during and/or analyzed during the current study are available from the corresponding author on reasonable request.

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
