# Peer review of "Characteristics of Collagen Changes in Small Intestine Anastomoses Induced by High-Frequency Electric Field Welding"

_biomolecules, 2022, doi:10.3390/biom12111683_

Round 1

Reviewer 1 Report

This document is very interesting because of its scientific approach to the qualification of a commercial device. However, it is difficult to read because it is based both on the results obtained and on the results of the literature without really bringing out the relationships. This is the case for the mechanical strength which is not studied in this article but which is mentioned several times. In the same way, the hypotheses on the quality of the suture which allows a good in-vivo reconstruction are not clearly stated. Furthermore, it is essential to provide one or more tables giving all the results obtained, both for collagen orientation and for Raman analysis, given the large number of conditions tested.

Specific points

1/ Indicate the number of Raman points carried out and the number of samples analyzed.

2/ In the discussion, delete or rewrite paragraph 2 (l238-l247) which is not supported by any data in this study.

3/ Include a specific discussion on the effect of the change in collagen cross-links and its conformation observed in Raman of long-term repair.

4/ Clearly state the acceptable limits in terms of pressure and temperature.

5/ line 253 “Better colcolation corresponds to better welding strength”, how is the welding strength determined in this study?

6/ This sentence "Collimation coefficient of collagen fibers is related to mechanical properties of tissues" is repeated twice in the text but is not supported by any data or reference. In addition, too much stiffness should lead to poor repair. Please specify the parameters and their maximum values from the present study that should result in a good quality of repair.

Author Response

Comments and Suggestions for Authors

This document is very interesting because of its scientific approach to the qualification of a commercial device. However, it is difficult to read because it is based both on the results obtained and on the results of the literature without really bringing out the relationships. This is the case for the mechanical strength which is not studied in this article but which is mentioned several times. In the same way, the hypotheses on the quality of the suture which allows a good in-vivo reconstruction are not clearly stated. Furthermore, it is essential to provide one or more tables giving all the results obtained, both for collagen orientation and for Raman analysis, given the large number of conditions tested.

Specific points

1/ Indicate the number of Raman points carried out and the number of samples analyzed.

Response 1: The mechanical strength in this paper refers to the burst pressure after the welding of the small intestine. I have changed the mechanical strength into burst pressure and added the data of burst pressure (Table1). Two tables (Table1 and Table2) were provided which respectively show collagen orientation and for Raman analysis. There were six samples fused at different Ps(n=3) and Ts(n=3) were imaged using Raman spectroscopy. Seven Raman points per sample were used for analysis.

2/ In the discussion, delete or rewrite paragraph 2 (l238-l247) which is not supported by any data in this study.

Response 2: I have made some major changes to this part, and in addition, I have added burst pressure data to further support. For details, please see the revised manuscript.

3/ Include a specific discussion on the effect of the change in collagen cross-links and its conformation observed in Raman of long-term repair.

Response 3: This paper mainly studies the changes of collagen in the process of in vitro tissue welding, and does not involve the changes of collagen in the process of long-term repair, which will be involved in future in vivo experiments.

4/ Clearly state the acceptable limits in terms of pressure and temperature.

Response 4: I have revised the draft to illustrate the pressure and temperature limits.The best pressure range is 150-250kpa, the highest temperature is best not to exceed 100℃, higher temperature means greater thermal damage to the surrounding tissue.

5/ line 253 “Better colcolation corresponds to better welding strength”, how is the welding strength determined in this study?

Response 5: I illustrate the relationship between collimation coefficient and welding strength in macroscopic and microscopic ways. The macroscopic method is to detect the welding strength of the fused tissue by burst pressure. The microscopic method is to examine the fusion quality by observing the tissue section at the fusion area with a microscope.

This paper is mainly determined by the size of the bursting pressure, and the supplementary detailed data can be found in Table1.

6/ This sentence "Collimation coefficient of collagen fibers is related to mechanical properties of tissues" is repeated twice in the text but is not supported by any data or reference. In addition, too much stiffness should lead to poor repair. Please specify the parameters and their maximum values from the present study that should result in a good quality of repair.

Response 6: Thank you very much for your reminding. There was a slight mistake in my previous description, and I have changed the relevant description. I have added relevant literature. It is mentioned in the reference of this paper that the colcolation coefficient of collagen is related to the mechanical characteristics of tissues, but the study on the mechanical characteristics of tissues is not the focus of this paper. The relationship between collimation coefficient of collagen fibers and burst pressure was studied in this study.

Too much pressure can lead to poor repair of the reason is that too much pressure can lead to tissue damage, the mechanical damage is not conducive to the late repair, and too much pressure will change the organization of physics parameters change, this includes the organization's conductivity, resistance and thermal conductivity, these parameters change will affect the change of temperature, the temperature cannot reach organization welding need to best temperature,Finally, it will lead to the failure of tissue welding.

According to the current research, a successful restoration requires appropriate pressure (150kpa-250kpa), power (110-140W), time (5-10S) and temperature (80-100℃). The quality of restoration is measured by the bursting pressure. As long as the bursting pressure is greater than 15.4mmHg, it can be regarded as a successful restoration. In this paper, the maximum bursting pressure is 59.8mmHg, which is greater than 15.4mmHg.

Reviewer 2 Report

The manuscript deals with the effect High-Frequency Electric Field on Collagen in Colonic Anastomoses by using ex vivo small intestine. This is an interesting topic and of relevance to the community. Overaal the manuscript is well written and the results are clear and convincing. I only have few comments:

Scanning electron microscopy images from normal and fused small intestine would improve the results about the collagen organization at the fibrillar level. It would be interesting to have more information about the samples preparation for each characterization technique.

Author Response

Comments and Suggestions for Authors

The manuscript deals with the effect High-Frequency Electric Field on Collagen in Colonic Anastomoses by using ex vivo small intestine. This is an interesting topic and of relevance to the community. Overaal the manuscript is well written and the results are clear and convincing. I only have few comments:

Scanning electron microscopy images from normal and fused small intestine would improve the results about the collagen organization at the fibrillar level. It would be interesting to have more information about the samples preparation for each characterization technique.

Response 1: I have provided electron microscopy images from normal and fused small fused area, please refer to Figure3 for detailed information. These figures provide the characteristics of the changes of collagen fibers in the small intestine after welding at the microscopic level. The main changes are density, morphology and distribution of collagen fibers.

Round 2

Reviewer 1 Report

All comments were taken into account in the revised version